# Influence of the Hot-Mix Asphalt Production Temperature on the Effectiveness of the Reclaimed Asphalt Rejuvenation Process

Edoardo Bocci [1],* , Emiliano Prosperi [2] and Maurizio Bocci [2]

1   Faculty of Engineering, Università eCampus, 22060 Novedrate, Italy
2   Department of Construction, Civil Engineering and Architecture, Università Politecnica delle Marche, 60131 Ancona, Italy
*   Correspondence: edoardo.bocci@uniecampus.it; Tel.: +39-338-467-7744

**Abstract:** Hot recycling of reclaimed asphalt pavement (RAP) into new hot-mix asphalt (HMA) is a complex process that must be precisely calibrated in the asphalt plants. In particular, temperature is a key parameter that, if inadequately set, can affect the final mix performance as it influences the RAP binder mobilization rate and the severity of bitumen short-term aging. The present paper aims at evaluating the effect of HMA production temperature on the behavior of mixtures including 50% of RAP and two types of rejuvenating agents. In particular, volumetric, mechanical, chemical, and rheological properties of the mixes and binder-aggregate adhesion have been investigated on the HMA produced in the laboratory at 140 °C or 170 °C. The results showed that the adoption of a lower production temperature did not significantly influence the air voids content in the mix, but determined a less stiff, brittle and cracking-prone behavior. Moreover, the decrease of the HMA production temperature was profitable for the increase of bitumen-aggregate adhesion.

**Keywords:** hot-mix asphalt; reclaimed asphalt; hot recycling; temperature; adhesion; FTIR; rheology





## 1. Introduction

Many studies and applications worldwide have demonstrated that the use of reclaimed asphalt pavement (RAP) in new hot-mix asphalt (HMA) has huge environmental and economic advantages. Thanks to the increasing attention to environmental issues, road administrations are currently having a greater broadmindedness towards hot recycling of RAP in terms of acceptance of higher RAP percentages in new HMA.

In the industrial production of HMA, the management of high percentages of RAP is challenging. The first issue is RAP heating. Different methodologies allow drying and heating RAP, for example parallel drums or recycling ring, but in many batch plants the RAP is still conveyed to the virgin aggregates in the elevator and is cold and partly wet when it reaches the mixing chamber [1]. Moreover, the adoption of high RAP contents determines a lower control of the mix gradation, especially if RAP is not crushed and separated into different fractions [2]. Indeed, hot recycling of RAP implies the use of rejuvenating agents (also called "rejuvenators"), which allow recovery of some of the chemical, rheological and mechanical properties that the RAP bitumen has lost with aging. However, rejuvenators should also be adequately managed by HMA producers in terms of type, dosage and addition location in the plant framework [3,4]. Rejuvenating agents have been the subject of several studies, which have highlighted their great benefits while also focusing attention on avoiding undesired effects. Table 1 summarizes the advantages and disadvantages of the rejuvenating agents.

**Table 1.** Summary of the benefits and disadvantages of rejuvenating agents.

| Benefits | References |
|---|---|
| Rejuvenators allow disrupting asphaltene clusters and restoring the volatiles components that bitumen lost with aging | [5,6] |
| Rejuvenators allow reducing binder stiffness and brittleness, enhancing fatigue and thermal cracking resistance | [7] |
| Rejuvenators can be produced from the processing of biological, secondary or waste materials | [8] |
| **Disadvantages** | **References** |
| Some additives only restore the bitumen oily components but do not act on the asphaltene clusters, reducing the rejuvenating effect | [9] |
| Rejuvenator dosage must be carefully defined to avoid the risk of mix cracking (low dosage) or rutting/moisture sensitivity (high dosage) | [10,11] |
| Rejuvenators must be homogeneously widespread in the bituminous phase to avoid localized distress on the pavement | [12] |
| Rejuvenators can lose their efficacy if subjected to high temperatures | [13] |

Within this complex system, a fundamental variable that greatly influences the production process and the final mix of volumetric and mechanical properties is the temperature. However, its importance is often neglected. During the HMA manufacturing, the temperature of the different components (bitumen, virgin aggregate, RAP, and rejuvenating agent) can vary in a large range [14].

On the one side, the adoption of higher temperatures for virgin aggregate and RAP determines a higher amount of RAP binder to melt and blend with the virgin bitumen [15]. However, the higher the temperature, the more harmful the short-term aging of the virgin and RAP bitumen [8,16]. Moreover, the greater mobilization of RAP bitumen entails a different virgin bitumen/RAP bitumen proportion, which may affect the properties of the aged-virgin bitumen blend [15,17].

On the other side, the use of lower aggregate and RAP temperatures at the plant has an opposite effect. In particular, a lower degree of blending between RAP and virgin bitumen is achieved with a lower penalization of the binder properties [18]. However, the minor heating of the HMA components determines a higher bitumen viscosity, which, in turn, may affect the mix compactability [19].

Some studies have recently investigated the effect of the production temperature on the effectiveness of the hot-recycling process, mainly focusing on warm-mix asphalt (WMA) techniques. The work by carried out by a RILEM (International Union of Laboratories and Experts in Construction Materials, Systems and Structures) inter-laboratory task group showed that a temperature decrease of 30 °C during the production of HMA including RAP determines an increased performance against rutting, cracking, and fatigue, without reducing the mix workability [20]. Another study demonstrated that some rejuvenating agents may suffer high temperatures and evaporate, partly or totally, reducing their effect on the restoration of RAP bitumen properties [13]. Carbonneau et al. obtained a comparable stiffness and fatigue performance between an HMA and a WMA (produced at a 28 °C lower temperature) including 40% of RAP content [21]. Yousefi et al. showed that WMA produced with 50% of RAP and a temperature reduction of 25 °C had higher fracture toughness and energy at low and intermediate temperatures compared to a reference HMA [22]. Rathore et al. reduced the HMA production temperature by 30 °C and included 60% RAP (without rejuvenating agents) but obtained a lower performance compared to the reference HMA without RAP [23].

Most of this research dealt with the characterization of bituminous binders, mastics or mixtures [24,25], but there are no studies on the interphase behavior with particular focus on the bitumen-aggregate adhesion. Some papers focused on the effect of the temperature on the adhesion [26–28], but they do not investigate the presence of RAP.

To have a clearer comprehension of the influence of temperature on the hot recycling of RAP, a global study, including the mix characterization and the evaluation of the adhesive properties between aggregate and binder, has been carried out. In particular, the objective of the present study is the evaluation of the influence of the HMA production temperature on the volumetric, mechanical, chemical, and rheological properties of the mixtures including 50% of RAP. Moreover, adhesion tests were carried out to assess the bond between bitumen (virgin and aged) and aggregate (virgin and RAP) at the different temperatures.

## 2. Materials and Methods

### 2.1. Experimental Program

To fulfil the research objective, HMA mixtures containing virgin aggregate, RAP, bitumen and rejuvenator were produced in the laboratory at 140 °C or 170 °C. Air voids content, voids filled with bitumen, indirect tensile stiffness modulus, indirect tensile strength, cracking tolerance index, and complex modulus were determined on gyratory compacted specimens. Fourier-transform infrared (FTIR) spectroscopic analysis was carried out on the bitumen extracted from the loose mixtures sampled before compaction. Moreover, binder bond strength (BBS) tests were carried out between virgin/recycled aggregate substrates and blends of aged and virgin bitumen with different proportions, by imposing the gluing temperatures of 140 °C and 170 °C. Specifically, the BBS tests were carried out with the aim to:

- Investigate the influence of the aged bitumen content on the binder adhesive properties;
- Evaluate which kind of aggregate, virgin or RAP, provides the higher adhesion with bitumen;
- Assess the influence of temperature on binder-aggregate adhesion.

Table 2 summarizes the experimental program.

**Table 2.** Experimental program.

| Property | Norm | Repetitions |
|---|---|---|
| Air voids content and voids filled with bitumen | EN 12697-8 | 4 |
| Indirect tensile stiffness modulus @ 20 °C | EN 12976-26 | 4 |
| Indirect tensile strength @ 25 °C | EN 12697-23 | 4 |
| Cracking tolerance index @ 25 °C | ASTM 8225-19 | 4 |
| Complex modulus @ $T_{ref}$ = 20 °C | AASHTO T342-22 | 2 |
| FTIR absorbance spectrum | - | 16 |
| Binder bond strength @ 25 °C | AASHTO TP-91 | 5 |

### 2.2. Materials

The HMA mixtures were designed to comply with Italian standards for binder layers with a maximum aggregate size of 16 mm. The mix gradation was obtained by blending 4 fractions of limestone virgin aggregates (coarse-grained and small-grained gravel, sand and filler) and 2 fractions of RAP. The coarse RAP was characterized by a particle diameter between 8 and 16 mm and a bitumen content of 4.8% by RAP weight, the fine RAP by a particle diameter lower than 8 mm and a bitumen content of 5.1% by RAP weight. The penetration at 25 °C and the softening point of the RAP bitumen were, respectively, $12 \times 10^{-1}$ mm and 77 °C. A fixed RAP content of 50% by aggregate weight was adopted for all the mixtures. Figure 1 shows the HMA gradation.

A 50/70 penetration bitumen, typically used in Italy to produce bituminous mixtures, was used as virgin binder. Table 3 shows the main physical and rheological properties of the bitumen. The dosage of virgin bitumen in the HMA was 2.7% by mix weight, which corresponded to a total bitumen content of about 5.2% by mix weight if considering the bitumen brought by the two RAP fractions (approximately 2.5% by mix weight).

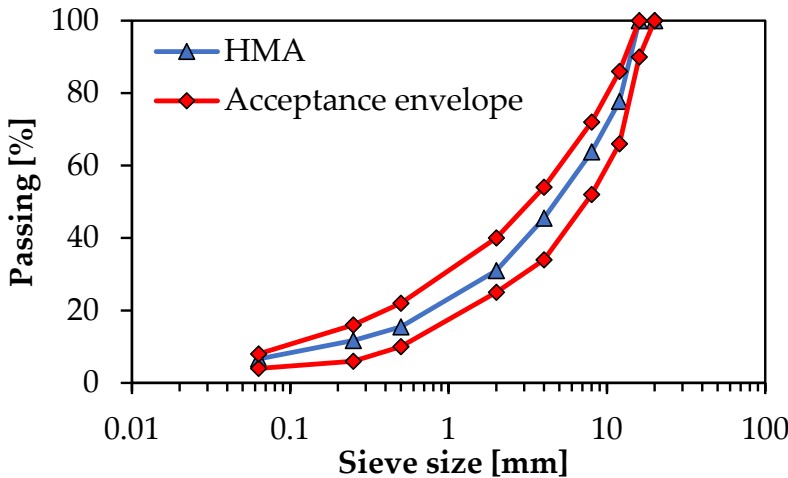

**Figure 1.** HMA gradation.

**Table 3.** Characteristics of the virgin bitumen.

| Property | Unit | Norm | Value |
|---|---|---|---|
| Penetration at T = 25 °C | $10^{-1}$ mm | EN 1426 | 55 |
| Softening point | °C | EN 1427 | 52.1 |
| Temperature G*/sinδ > 1 kPa | °C | EN 14770 | 64 |
| Glover-Rowe parameter at T = 15 °C, ω = 0.005 rad/s | Pa | - | 189 |

Two commercial rejuvenating agents were used in the mixtures: rejuvenator A consists of modified polyamines and vegetal oils, while rejuvenator B is a crude tall oil derived from pine wood and includes resin acids, fatty acids, and unsaponifiable. The rejuvenating agents were analyzed through FTIR spectroscopy. Table 4 summarizes the characteristic peaks of the two products. According to the results of a previous study [29], the rejuvenator dosages were 9% and 6% by RAP binder weight respectively for additives A and B.

**Table 4.** Characteristic peaks of the rejuvenators from FTIR analysis.

| Wavenumber | Compound | Group | Rej. A | Rej. B |
|---|---|---|---|---|
| 3010 cm$^{-1}$ | Alkene | C-H stretching | High | Medium |
| 1742 cm$^{-1}$ | Ester | C=O stretching | Small | High |
| 1589 cm$^{-1}$ | Amine | N-H bending | High | Absent |

The HMA laboratory production provided the heating of the mix components in the oven at 140 °C or 170 °C: the virgin aggregate and the RAP were heated for 3 h, the virgin bitumen for 2 h. The rejuvenating agents were added to the virgin bitumen 30 min before HMA mixing. The HMA mixing was carried out using an automatic mixer and including, in order, coarse aggregate, RAP, virgin bitumen and filler, according to EN 12697-35. The loose mix was kept in the oven for 30 min, and afterwards, cylindrical specimens were produced using a gyratory compactor with angle 1.25°, vertical pressure 600 kPa and 100 gyrations, according to EN 12697-31.

### 2.3. Test Protocols

The bulk density was obtained by weighing the specimen in air and in water, without applying any sealing, according to EN 12697-6 procedure A. The maximum density was calculated from aggregate and bitumen densities and proportion (EN 12697-5, procedure C). The air voids content (AVC) of the specimens was determined as the complement to 1 of the ratio between bulk and maximum densities, in percentage. Moreover, the content of

voids filled with bitumen (VFB) was calculated as the percentage of void in the aggregate skeleton occupied by the binder.

The Indirect Tensile Stiffness Modulus (ITSM) tests were carried out using a servo-pneumatic device, according to EN 12697-26-Annex C. Pulse loads with a rise time of 0.124 s were applied to reach the target horizontal deformation of 5 µm. The ITSM was calculated with the following equation:

$$\text{ITSM} = \frac{F \cdot (v + 0.27)}{z \cdot h} \tag{1}$$

where $F$ (N) is the vertical load peak; $v$ is the Poisson's ratio (0.35); $z$ (mm) is the horizontal deformation and $h$ (mm) is the mean specimen height. The test was performed at 20 °C and provided 4 repetitions under the same conditions.

An electro-mechanical press allowed performing the Indirect Tensile Strength (ITS) tests at 25 °C according to EN 12697-23. The test provided a constant deformation rate of 50 ± 2 mm/min until the load reached, after the failure, a value equal to 30% of the peak. From the vertical load versus vertical displacement curves, the Cracking Tolerance Index (CTI) was calculated according to ASTM 8225-19. CTI is an indicator of the specimen ductility and can be calculated through the following equation:

$$\text{CTI} = \frac{h}{h_{ref}} \cdot \frac{G_f}{m_{75}} \cdot \frac{l_{75}}{d} \tag{2}$$

where $h$ (mm) and $d$ (mm) are the mean specimen thickness and diameter, respectively; $h_{ref}$ (mm) is the reference thickness equal to 62 mm; $G_f$ (N/mm) is the fracture energy (i.e., the area of the load-displacement curve divided by the specimen section $h \cdot d$); $l_{75}$ (mm) and $m_{75}$ (N/mm) are respectively the displacement and the slope of the load-displacement curve when the load decreases to 75% of the peak. ITS and CTI were determined as the average on 4 test repetitions.

The linear viscoelastic (LVE) characterization of the mixtures was carried out through uniaxial cyclic compression tests, according to AASHTO T378-17. In particular, the complex modulus $E^*$ of the specimens was measured using a servo-hydraulic universal testing machine. During the test, haversine compression loads were applied to achieve a vertical strain amplitude of 50 microstrain ($50 \times 10^{-6}$ mm/mm). The test was performed at four temperatures (5 °C, 20 °C, 35 °C and 50 °C) and eight frequencies (from 0.1 to 20 Hz). Two specimens were tested for each mixture.

A hot extractor (wire mesh filter) with trichloroethylene and a rotary evaporator were used to extract the bitumen from the mixtures, according to EN 12697-1 and EN 12697-3. The bitumen was analyzed through FTIR spectroscopy in Attenuated Total Reflectance (ATR) mode with a diamond crystal. The spectra were captured within a wavenumber range between 4000 and 600 cm$^{-1}$, with a resolution of 4 cm$^{-1}$. 16 spectra were accumulated to obtain the final binder spectrum. Carbonyl ($I_{CO}$) and Sulfoxide ($I_{SO}$) Indexes were calculated from the FTIR spectra to quantify the aging effects. According to [30], $I_{CO}$ and $I_{SO}$ were calculated using the following equations:

$$I_{CO} = \frac{A_{1690}}{A_{1460} + A_{1375}} \tag{3}$$

$$I_{SO} = \frac{A_{1030}}{A_{1460} + A_{1375}} \tag{4}$$

where $A_{1690}$ is the area of the spectrum around the carbonyl group (1690 cm$^{-1}$); $A_{1030}$ is the area of the spectrum around the sulfoxide group (1030 cm$^{-1}$); $A_{1460}$ and $A_{1375}$ are the reference areas of the ethylene (1460 cm$^{-1}$) and methyl (1375 cm$^{-1}$) peaks.

A self-aligning Pneumatic Adhesion Tensile Testing Instrument (PATTI) equipment was used to perform binder bond strength (BBS) tests, according to AASHTO TP-91. Two substrates, reproducing virgin limestone aggregate and RAP, three RAP/virgin binder rates (20/80, 35/65 and 50/50), two rejuvenators (coded with the letters A and B) and two

bitumen application temperatures (140 °C and 170 °C) were investigated. In particular, the RAP substrate was simulated by spreading about 10 µm of virgin bitumen at 170 °C on hot limestone plates (at the same temperature). According to AASHTO R30, the plates underwent two conditions in the oven, respectively at 135 °C for 4 h and at 85 °C for 120 h. The RAP bitumen was produced in the laboratory from the virgin bitumen by aging it according to RTFOT (85 min at 163 °C) and PAV (20 h at 100 °C and 2.1 MPa) protocols. The RAP and virgin bitumens were mixed with proportions 20/80, 35/65 and 50/50 to evaluate the effect of different mobilization rates of the RAP bitumen. The rejuvenating agents (A or B) were added to the binder during the blending of RAP and virgin bitumens. The temperature of 140 °C or 170 °C was imposed during RAP/virgin binder stirring. To glue the pull stub on the plates, these were heated in the oven for 2 h, the limestone plates at 140 °C or 170 °C, the RAP plates at 60 °C. Little bitumen "balls" were positioned on the stubs at room temperature; then, they were heated at 140 °C or 170 °C for 30 min and finally glued on the plates. The BBS tests were carried out at 25 °C, and 5 repetitions were provided.

The flowchart in Figure 2 summarizes the experimental program and the specimen preparation methods.

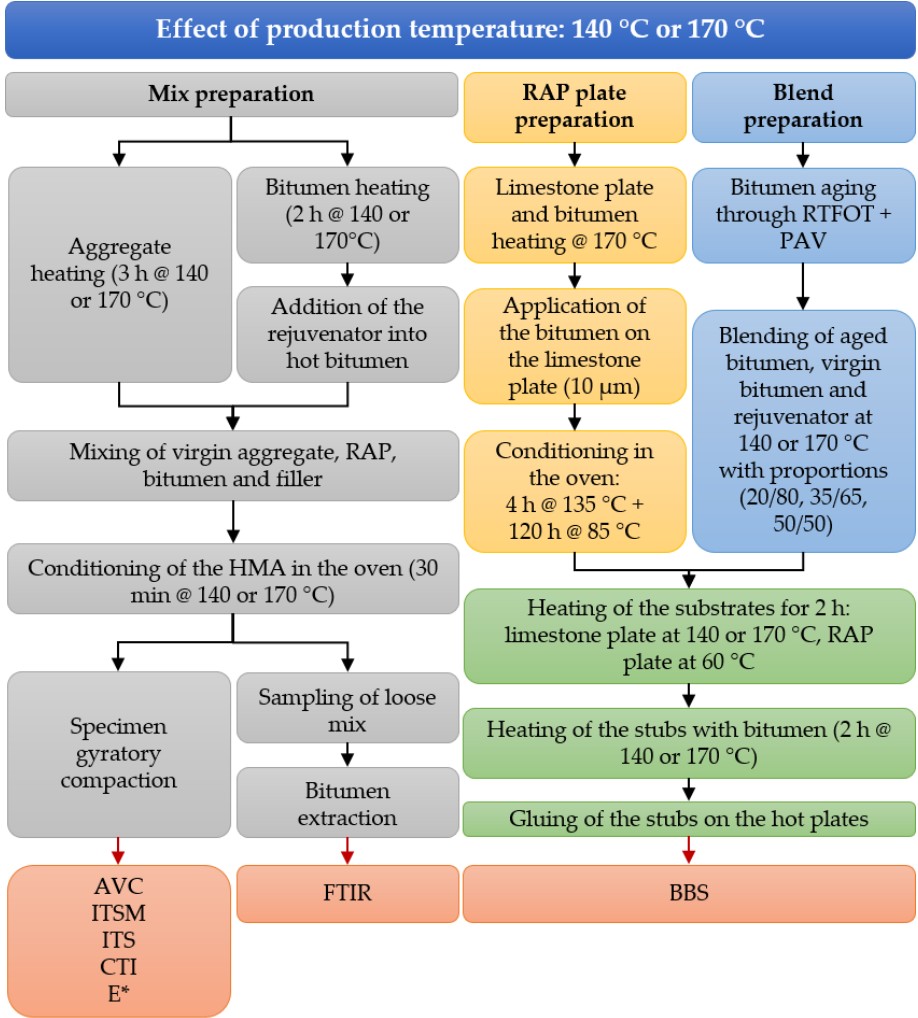

**Figure 2.** Experimental program and specimen preparation.

## 3. Results

### 3.1. Volumetric Properties

Figure 3 shows the volumetric properties of the mixtures in terms of air voids content (AVC) and voids filled with bitumen (VFB). It can be observed that the volume of void in the

specimens was rather low (ranging between 2 and 4%) for all the HMA mixes due to high values of VFB (>75%). Despite a certain data scattering, the volumetric properties seemed to be independent from the HMA production and compaction temperature (140 °C or 170 °C) and from the rejuvenator type (A or B). This denoted that the mixtures were adequately compactable even at 140 °C due to a correct estimation of the virgin bitumen content.

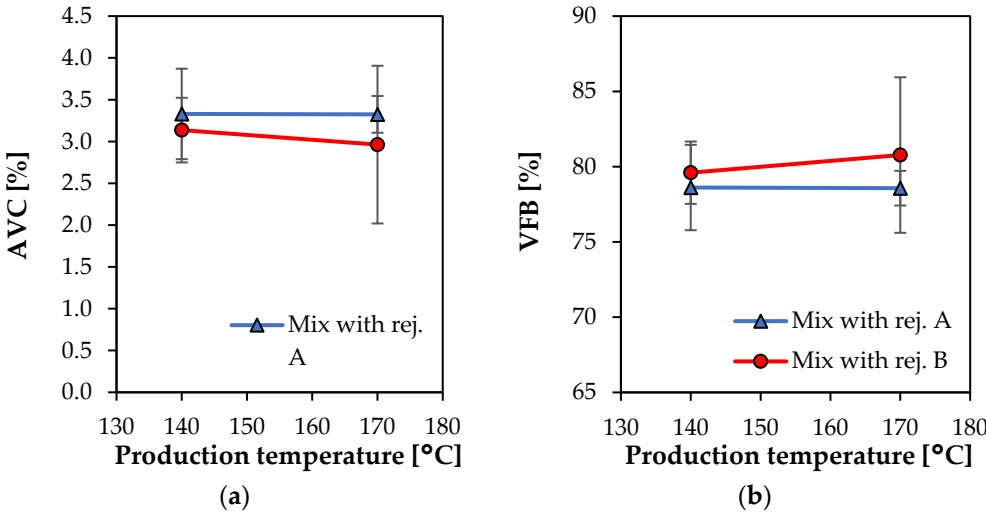

(**a**)　　　　　　　　　　　　　　　　　　　　(**b**)

**Figure 3.** Volumetric properties of the mixtures: (**a**) Air voids content; (**b**) Voids filled with bitumen.

### 3.2. Indirect Tensile Stiffness Modulus

Figure 4 depicts the values of ITSM measured at 20 °C. It can be immediately noted that the HMA stiffness was noticeably influenced by the production temperature. An increase of the temperature from 140 °C to 170 °C determined an increase of ITSM by 47% and 40%, respectively, for the HMA with rejuvenator A and B. In particular, the stiffness of the mix with rejuvenator A increased from 7600 to 11,100 MPa; for the mix with rejuvenator B, the ITSM rose from 9800 to 13,700 MPa.

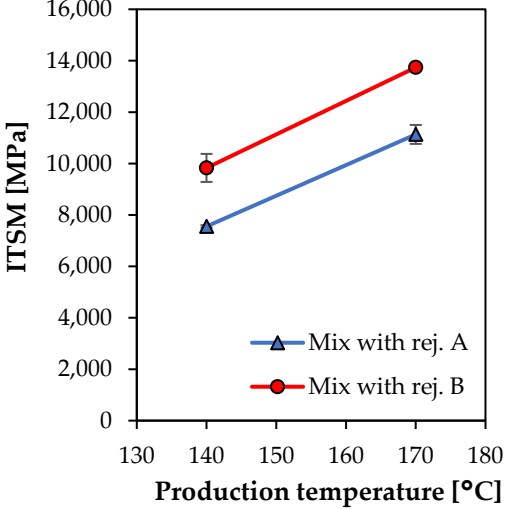

**Figure 4.** Variation of ITSM as a function of mix production temperature and rejuvenator type.

This result was related to the more severe short-term aging that the virgin bitumen and the aged RAP bitumen achieved from HMA production to specimen compaction at the higher temperature of 170 °C. Moreover, a higher degree of blending between virgin and RAP bitumen probably happened when mix production temperature was 170 °C. Among the rejuvenating agents, additive A allowed a greater reduction of the RAP bitumen stiffness, thus a lower ITSM of the mixture, compared to additive B.

### 3.3. Indirect Tensile Strength Test

Figure 5 shows the results from indirect tensile tests in terms of ITS and CTI. In can be observed that ITS values were higher at the production temperature of 170 °C for both the HMA mixtures with rejuvenator A or B. In particular, ITS increased from 1.24 MPa to 1.46 MPa (+18%) when using rejuvenator A, from 1.41 MPa to 1.78 MPa (+26%) when using rejuvenator B. It is highlighted that only the HMA with rejuvenator A produced at 140 °C complied with Italian national specifications [31], which provided a maximum ITS of 1.40 MPa.

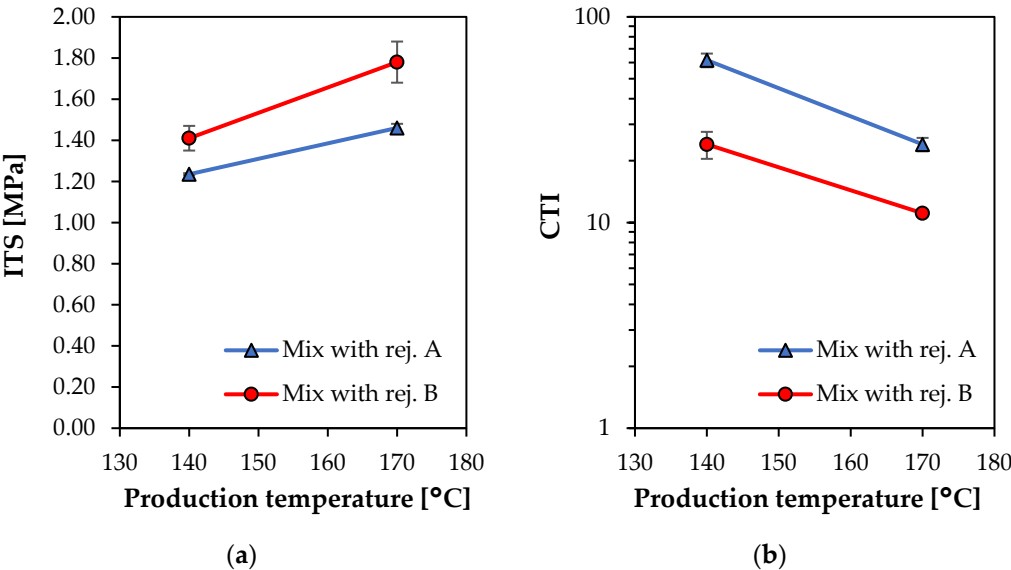

**Figure 5.** Variation of ITS (**a**) and CTI (**b**) as a function of mix production temperature and rejuvenator type.

The CTI, which represents the ability of the HMA to contrast crack propagation, showed a decreasing trend when increasing the mix production temperature. For both the mixes with rejuvenator A and B, the higher production temperature corresponded to a higher brittleness, indicated by a decrease of CTI of about 60%.

As in the case of the ITSM, the results from indirect tensile tests confirmed that the adoption of high production temperature significantly affected the HMA performance due to the amplified short-term aging and to the mobilization of a greater amount of RAP bitumen, which blended with the virgin one into a harder and stiffer binder. Again, the rejuvenator A showed a higher effectiveness mainly related to the higher dosage.

### 3.4. FTIR Spectroscopic Analysis

Figures 6 and 7 show the results from FTIR spectroscopy on the bitumens extracted from the different HMA mixes. Specifically, Figure 6 depicts the FTIR spectra between the wavenumbers of 2000 and 600 cm$^{-1}$. It can be observed that all the binders presented clear bands at 1690 cm$^{-1}$ and 1030 cm$^{-1}$, denoting the presence of the carbonyl (C=O) and sulfoxide (S=O) groups generated by oxidation. Moreover, the bitumen extracted from the HMA mixes produced at 170 °C showed a higher height of these bands, particularly the one at 1690 cm$^{-1}$. During the bitumen extraction a total recovery of RAP binder and a full blending with the virgin one happened independently from the mix production temperature. Therefore, the taller peaks observed for the HMA produced at 170 °C was clearly an effect of the more severe short-term aging of the virgin and RAP bitumens. The increasing content of the oxidation products when adopting a high HMA production temperature was quantified in terms of carbonyl index $I_{CO}$ and sulfoxide index $I_{SO}$ (Figure 7). In particular, the $I_{CO}$ increased by 82% and 60%, respectively, when rejuvenator A and B were used, confirming what was visually observed in the FTIR spectra. Differently, the increase of $I_{SO}$

when raising the mix production temperature was significant in the case of rejuvenator A (+32%), while only a 2% increase was determined for the HMA with rejuvenator B.

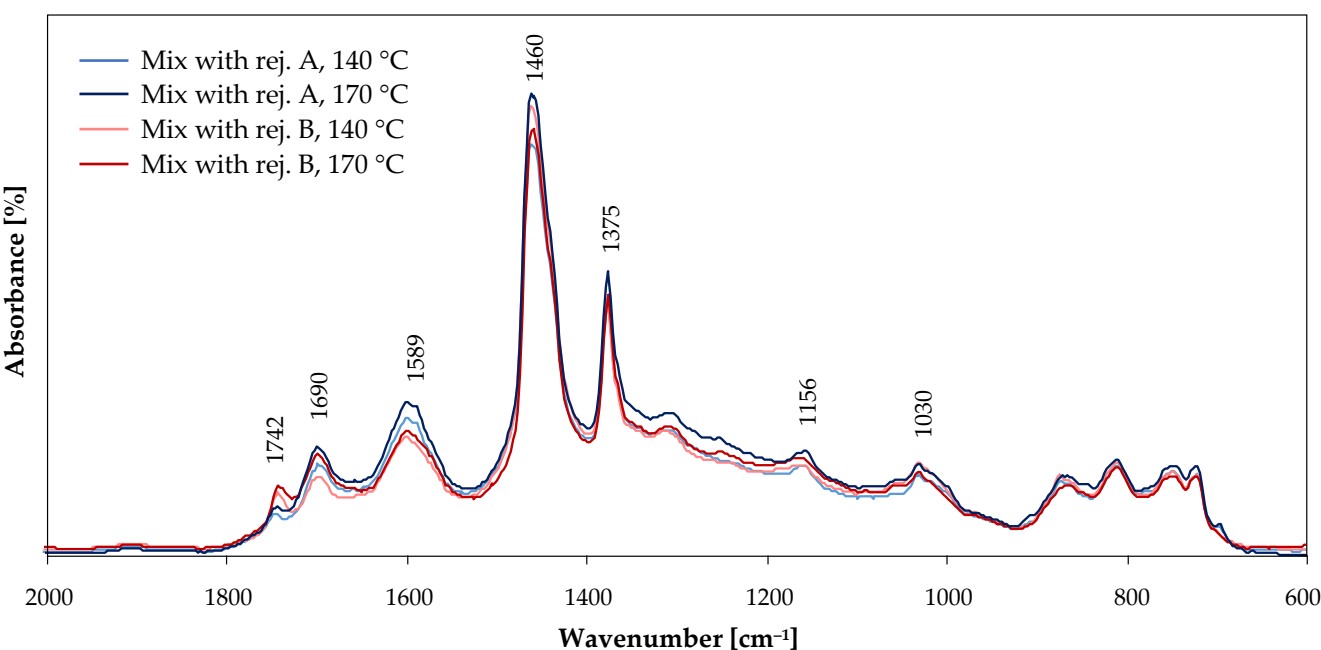

**Figure 6.** Absorbance spectra from FTIR analysis on the extracted binders.

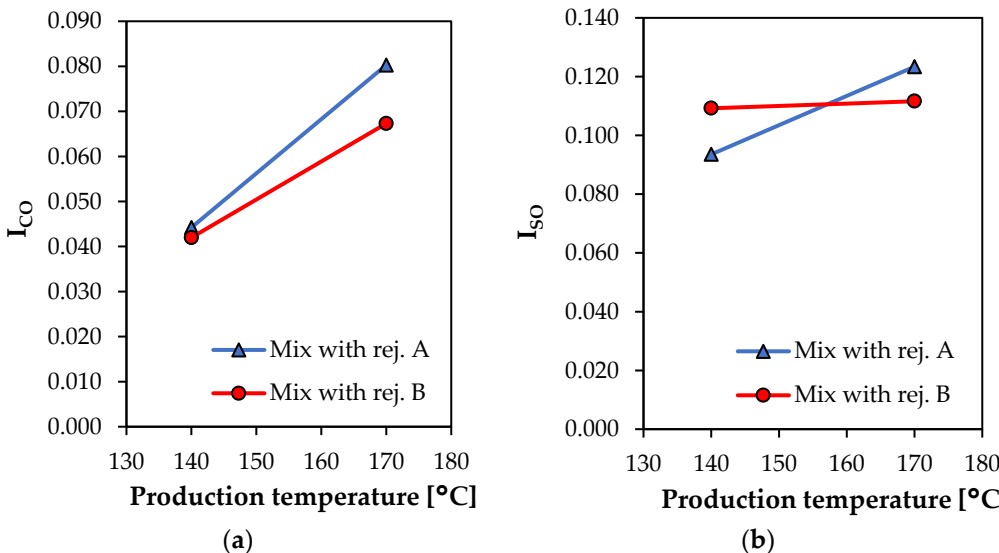

**Figure 7.** Oxidation indexes from FTIR analysis: (**a**) Carbonyl index $I_{CO}$; (**b**) Sulfoxide index $I_{SO}$.

Finally, it was observed that the FTIR analysis allowed tracing the presence of the rejuvenators by analyzing their characteristic bands. In particular, the band at 1742 cm$^{-1}$ was evident in the spectra of all the binders (Figure 6), especially in those from the HMA with rejuvenator B. Similarly, the presence of amines in rejuvenator A determined a higher height of the peaks at 1589 cm$^{-1}$ in the bitumen from the HMA mixes where this additive was used. This result supports what was hypothesized in previous studies [32], i.e., that FTIR spectroscopy can allow evaluating the presence and even the dosage of rejuvenators in HMA including hot-recycled RAP.

### 3.5. LVE Characterization

The measured rheological data (norm and phase angle of the complex modulus, storage and loss moduli) were depicted in the Black space and Cole-Cole plot, as shown in Figure 8. From these graphs, it can be noted that the complex modulus $|E^*|$ approximately varied between 300 MPa and 22,000 MPa, while the phase angle $\varphi$ varied between 5° and 32°. The experimental data in Black and Cole–Cole diagrams aligned along a regular trend, confirming the validity of the time-temperature superposition principle (TTSP). This allowed considering the materials as thermo-rheologically simple and applying the temperature shift factors to build the complex modulus and phase angle master curves. In particular, the measured data were shifted with respect to time to obtain single isothermal functions at the reference temperature (20 °C).

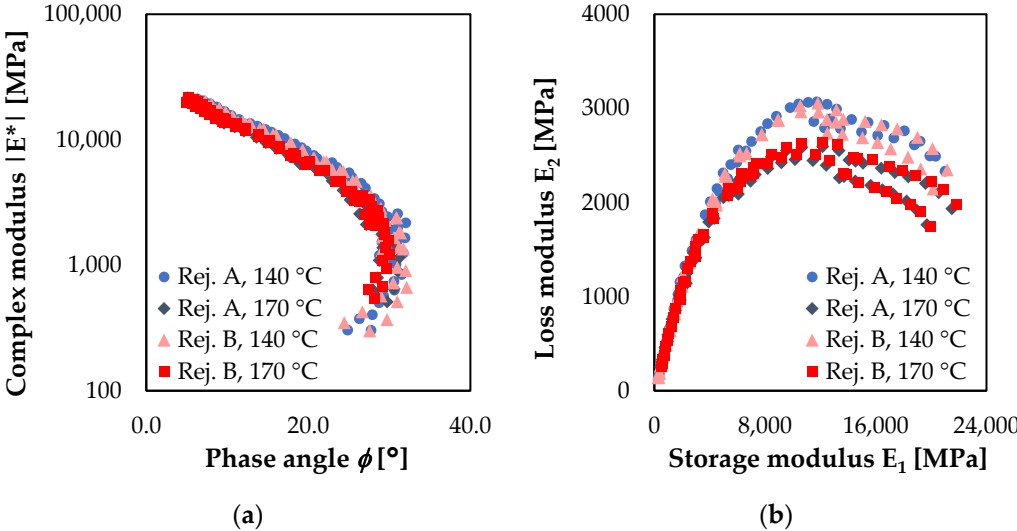

(**a**)                       (**b**)

**Figure 8.** Rheological data: (**a**) Black space; (**b**) Cole–Cole plot.

The Generalized Logistic Sigmoidal (GLS) model was adopted to represent the experimental data [33,34]. The definition of the norm of the complex modulus and the phase angle as a function of the reduced frequency $f_r$ are shown in the following equations:

$$\log E^*(f_r) = \delta + \frac{\alpha}{\left[1 + \lambda e^{[\beta+\gamma(\log f_r)]}\right]^{1/\lambda}} \tag{5}$$

$$\phi(f_r) = -90\alpha\gamma \frac{e^{[\beta+\gamma(\log f_r)]}}{\left[1 + \lambda e^{[\beta+\gamma(\log f_r)]}\right]^{(1+1/\lambda)}} \tag{6}$$

where $\log f_r$ is the logarithm of the reduced frequency, $\delta$ is the static asymptote, $\alpha$ is the difference between the values of the glassy and static asymptote, $\lambda$, $\beta$ and $\gamma$ are shape parameters.

Figure 9 shows the master curves of $|E^*|$ and $\varphi$ at the reference temperature of 20 °C. The results showed that the mixtures produced at 140 °C had a lower stiffness compared to the HMA produced at 170 °C (Figure 9a). In particular, the difference was higher at low frequencies/high temperatures, while comparable $|E^*|$ values (approximately 21,000 MPa) were obtained for all the mixtures at high frequencies/low temperatures. This denoted that at low temperatures the binder was equally stiff, and the different short-term aging due to the different HMA production temperature was not appreciable. Differently, at high temperatures, the more aged binder of the mixtures produced at 170 °C opposed softening and entailed a stiffer specimen behavior.

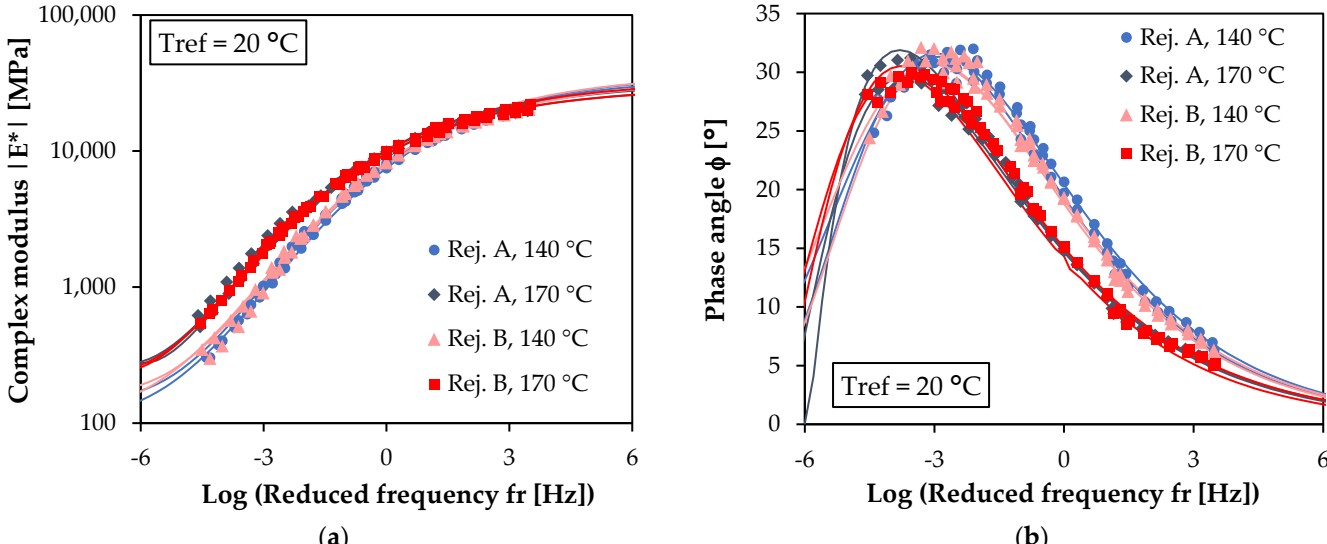

**Figure 9.** Master curves of the tested specimens: (**a**) Complex modulus |E*|; (**b**) Phase angle $\varphi$.

The HMA production temperature also influenced the phase angle values (Figure 9b). In general, the mix produced at 170 °C showed lower phase angles compared to the mix produced at 140 °C. The difference was maximum in the middle of the reduced frequency spectrum (from 0.01 to 100 Hz), where it reached about 6°. This indicated a more pronounced viscous behavior of the mixes produced at 140 °C, which reflected a higher ability to relax the stresses induced by traffic loading and delay the formation of cracks.

When comparing the |E*| and $\varphi$ master curves of the HMA including different rejuvenators, a similar behavior was observed for the same mix production temperature. For all the tested specimens, a good superposition of the experimental data with the GLS model function was achieved.

### 3.6. Binder-Aggregate Adhesion

The results from the BBS tests are shown in Figure 10 in terms of pull-off tensile strength (POTS). It can be observed that the adhesive properties were inversely proportional to the content of aged bitumen in the binder blend. This result was obtained for each type of rejuvenator, HMA production temperature and substrate (virgin and RAP). As the POTS values were slightly scattered, the statistical analysis of the experimental results through *t*-test was carried out. In particular, the populations of POTS values were considered statistically different if the probability $\alpha$ was lower than 0.05 (i.e., 5%).

The comparison of the POTS measured for the bituminous blends including 20% and 35% of RAP bitumen provided a value of $\alpha$ of $1.2 \times 10^{-6}$. Similarly, $\alpha$ was $6.1 \times 10^{-7}$ when comparing the blends with 35% and 50% of RAP bitumen. So, the statistically analysis confirmed that the binder-aggregate adhesion decreased when increasing the aged bitumen content.

Among the different substrates, the adhesion was slightly higher on the limestone plates for the RAP bitumen content of 20%, with $\alpha$ that was a little lower (0.03) than the limit. On the contrary, for the higher RAP bitumen contents (35% and 50%), the POTS were greater for the RAP substrate than for the limestone ($\alpha = 8.4 \times 10^{-5}$). Probably, the adhesion on the virgin aggregate was more significantly influenced by the binder stiffness, so that the POTS were noticeably higher for the 20/80 RAP/virgin bitumen proportion but noticeably decreased when raising the RAP bitumen content. Differently, on the RAP substrate the decrease of the adhesion with the RAP bitumen content was less important.

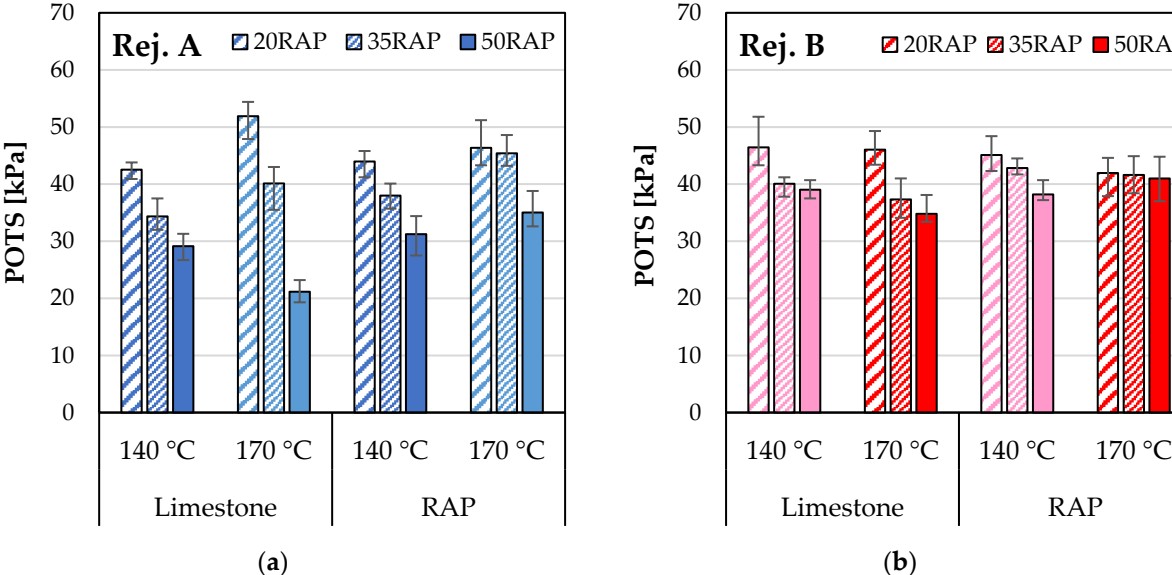

**Figure 10.** Pull-off tensile strength (POTS) values: (**a**) bituminous blends with rejuvenator A; (**b**) bituminous blends with rejuvenator B.

When comparing the effect of the two rejuvenators, it can be noted that the POTS values were similar for the RAP/virgin bitumen proportions of 20/80 ($\alpha$ = 0.33) and 35/65 ($\alpha$ = 0.37). For the RAP bitumen content of 50%, rejuvenator B allowed obtaining a higher adhesion, as statistically supported by $\alpha = 1.2 \times 10^{-6}$.

From the graphs, it can also be noted that the bitumen application temperature (140 °C or 170 °C) did not show a great influence on the adhesive properties. The statistical analysis confirmed this result, as the $\alpha$ value was higher than the limit ($\alpha$ = 0.50). This denoted that the more severe aging achieved by the binder at high temperature counterbalanced the higher adhesiveness.

## 4. Discussion

The study confirmed that temperature is an important variable that must be precisely managed during the HMA production, especially when RAP is used. In many asphalt plants, virgin aggregates are often overheated to ensure an adequate transfer of heat to the RAP particles and avoid mix workability issues. However, inaccuracies on the aggregate conditions (temperature and humidity) and worries about the risk of laying a too cold HMA can lead to an excessively high mix temperature, which is not necessary or even detrimental for an effective compaction. The volumetric analysis on the laboratory-produced HMA mixes including 50% of RAP demonstrated that, in the specific context of the present research, the material was easily compactable at both 140 °C and 170 °C (Figure 2).

If the different mix temperature did not influence the specimen volumetrics, it noticeably affected the mechanical, chemical and rheological behavior. In particular, the higher HMA temperature entailed two phenomena:

1.   the heavier short-term aging of the binder, especially the virgin bitumen [4], and
2.   the mobilization of a greater amount of RAP bitumen, which resulted in a higher aged/virgin bitumen ratio in the effective binder phase (the non-melted RAP bitumen kept on covering the old aggregate particles and, only in a very small part, diffused in the new bitumen).

Consequently, the HMA binder became more oxidized (as observed from FTIR analysis) and the mix showed a stiff, brittle and cracking-prone behavior (higher ITSM and ITS, lower CTI, higher $|E^*|$ and lower $\varphi$).

Regarding the influence of HMA production temperature on the bitumen-aggregate bond, the bitumen should theoretically increase its binding ability as it reduces the viscosity

when it is heated to higher temperature. On the other side, aging and oxidation tend to make the bitumen harder but less adhesive. In the BBS tests carried out in the present study, these effects reciprocally compensated, so that the temperature at which the binder was stuck to the aggregate did not influence the POTS (Figure 9). However, the variable that most affected the adhesion was the RAP/virgin bitumen ratio, which increased when increasing the HMA production temperature. Therefore, the reduction of the mixing temperature as it implies the lower mobilization of the RAP binder is positive for the increase of adhesion between the bitumen and aggregate, both in the case of virgin and pre-coated RAP particles.

## 5. Conclusions

The present research dealt with the evaluation of how the reduction of the HMA production temperature from 170 °C to 140 °C influences the volumetric, mechanical, chemical, rheological, and adhesive properties of the mixtures including 50% of RAP and two different rejuvenators. In light of the experimental results, the following conclusions can be drawn:

- The HMA production temperature did not influence the air voids content and voids filled with bitumen of the specimens prepared with gyratory compaction;
- An increase of the temperature from 140 °C to 170 °C determined an increase of stiffness (higher ITSM of about 45%) and strength (higher ITS of about 20%) and a reduction of the ductility (lower CTI of about 60%);
- The FTIR spectroscopy allowed observing a higher amount of the oxidation products (especially the carbonyl groups) in the chemical structure of the bitumen extracted from the specimens produced at 170 °C;
- The LVE characterization showed that the mix produced at 170 °C had higher stiffness, especially at high temperatures, and lower ability to relax stress, related to the lower viscous properties;
- The binder-aggregate adhesion was noticeably affected by the ratio between RAP bitumen and virgin bitumen in the binder blend (the higher this ratio, the lower the adhesion). So, the adoption of lower HMA production temperatures can increase the bond between the bitumen and aggregate, both in the case of virgin and the pre-coated RAP particles, as it determines a lower mobilization of the RAP binder.

The laboratory tests highlighted that the reduction of the HMA production temperature is evidently positive for the performance of the mixtures including RAP. The promising findings encourage further research on this topic. In particular, the main limitation of this study lies in the specimen production process in the laboratory, which can present some differences from the plant production. Moreover, in the present research only two HMA production temperatures, 140 °C and 170 °C, were considered. For this reason, future investigations will regard other production temperatures and a real scale field application. This can also include the possibility of using warm mix asphalt solutions to further reduce the HMA manufacturing temperature. These technologies would certainly allow the decrease of the fume and pollutant emissions during HMA production and laying, which is fundamental in the optics of more and more sustainable roads, but the effect on the mix behavior and bitumen-aggregate adhesion still need to be explored.

**Author Contributions:** Conceptualization, E.B. and M.B.; methodology, E.B.; formal analysis, E.P.; investigation, E.P.; data curation, E.B.; writing—original draft preparation, E.B.; writing—review and editing, E.P.; supervision, M.B. All authors have read and agreed to the published version of the manuscript.

**Funding:** This research received no external funding.

**Data Availability Statement:** Not applicable.

**Acknowledgments:** The authors greatly thank InCoBit S.r.l. for providing the raw materials (aggregate, RAP, virgin bitumen and additives) used in this study.

**Conflicts of Interest:** The authors declare no conflict of interest.

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
