# Peer review of "Influence of the Hot-Mix Asphalt Production Temperature on the Effectiveness of the Reclaimed Asphalt Rejuvenation Process"

_infrastructures, doi:10.3390/infrastructures8010008_

Round 1
Reviewer 1 Report
Dear Author,
You did a good job. This is an interesting topic regarding the application of RAP.
Please consider the attached PDF file, including comments.
Thank you.

Reviewer 2 Report
The paper is well-written, suits the journal’s aim and it is believed that the study is of some value for both researchers and practitioners. It has enough experimental testing and solid results. I have the following comments for the authors for further improvements:
1_ Lines 37-39, 61-63: Please elaborate more on the pros and cons of rejuvenation. Summarizing remarks within a table could be informative for the readership.
2_ The literature review is sufficient but the objective statement in lines 75-77 is missing. Please revise and make it clearer, e.g., the purpose of this research study is to investigate/perform a holistic investigation…
3_ Line 79: the title should be kept as “experimental program”.
4_ Add units for all quantities in all equations.
5_ Line 57: Which protocol did you follow for the uniaxial compression tests?
6_ I would suggest adding a flowchart to accompany the description given in section 2.
7_ Lines 308-314: Please try to elaborate on the objective statement by including phrasing from these lines (recall remarks No. 1 and 5) and keep the text herein simpler.
8_ Lines 350-351: Did you perform any kind of statistical testing to demonstrate the importance of temperature. Are the four replicates enough? Please comment and consider adding study limitations (if any).
9_ Lines 371-372: You might avoid questions herein. Please reconsider.
Reviewer 3 Report
While the study attempts to find a very interesting aspect, the very fundamental question the whole argument line on is – Is the viscosity within the range of our interest during the mix and compaction? It is recommended that for asphalt binders, the laboratory mixing and compaction temperatures should be determined where the viscosity‐temperature line crosses the viscosity ranges of 0.17 ± 0.02 Pa‐s (mixing temperature range) and 0.28 ± 0.03 Pa‐s (compaction temperature range). The temperature for the desired viscosity is determined following a recommended procedure (Usually taking the viscosity at different temperatures and drawing a line as mentioned above). The recommended viscosity range is a prerequisite to selecting a specific compaction temperature. So, the conclusion observed in the study requires a cross-check that the viscosity of the asphalt lies within the range for the selected temperatures (140 0C to 170 0C ). As the study did not use any additive (WMA or other additives), I think the range could be narrower than the selected range (300C ). To be precise, temperatures that result in the viscosity of 0.28 ± 0.03 Pa‐s are expected to be the scope of the study; probably, the author could argue about the need for the study then.
- Lines 122-128 require standard protocol.
- Figure 2, I encourage the authors to cross-check the result, as it seems to be a little bit away from the expectation for such a wide range of temperatures. Also, Figure 2 and Figure 3 contradict each other as the lower void is expected to have such a trend in figure 3, which is not the case with figure 2,
- The authors have concluded different aspects which merely direct to have an integrated conclusion.
Round 2
Reviewer 2 Report
The paper was improved and the reviewer's concerns were replied. Some additional comments:
1_ Line 41: Please replace pros and cons with a more formal wording, e.g., advantages and disadvantages.
2_ Lines 453-455: Please include a short comment from a sustainable point of view about the emission reductions from WMA technology.
Reviewer 3 Report
The authors have made improvements and could be considered for publication.
Author Response
Thank you for your effort in the paper review process.
Round 3
Reviewer 2 Report
There are no further comments.